# Externalities in wild pig damages on U.S. crop and livestock farms: The role of landowner actions and landscape heterogeneity

**Sophie C. McKee**[1,2,⊕,¤a,*], **Nathan D. DeLay**[3,⊕,¤b], **Daniel F. Mooney**[3,⊕,¤b], **Stephanie A. Shwiff**[1,¤a]

**1** National Wildlife Research Center, United States of America Department of Agriculture, Animal and Plant Health Inspection Service, Wildlife Services, Fort Collins, Colorado, United States of America,
**2** Department of Economics, Colorado State University, Fort Collins, Colorado, United States of America,
**3** Department of Agricultural and Resource Economics, Colorado State University, Fort Collins, Colorado, United States of America

¤a Current address: National Wildlife Research Center, U.S. Department of Agriculture, Animal and Plant Health Inspection Service, Wildlife Services, 401 Laporte Ave, Fort Collins, Colorado, 80521, United States of America
¤b Current address: Colorado State University, Department of Agricultural and Resource Economics, 501 University Avenue, Fort Collins, CO 80521, United States of America
* sophie.mckee@colostate.edu
⊕ These authors contributed equally to this work.

## Abstract

Invasive wild pigs can impose significant economic costs on crop and livestock farms. Many factors influence the incidence and intensity of these losses, making efforts to reduce or eradicate these populations complex. While farm and ranch operators may perceive wild pigs as agricultural pests, other landowners often see them as wild game with recreational value. This study investigates the relationship between landowner practices that attract wild pigs and the likelihood of pig presence and damage on farm and ranch operations. It considers the farmers' own actions that attract wildlife, neighboring landowner actions, the heterogeneity of the surrounding landscape, and county-level factors. The findings show a significant and positive associations between neighbors' actions and the probability of wild pig presence and financial losses from wild pig damage. Additionally, increasingly heterogeneous landscapes may further exacerbate this challenge. This research indicates that the choices made by adjacent property owners can undermine the effectiveness of public and private efforts to manage wild pig populations. Conversely, the impacts of wild pig management likely extend beyond specific management areas. Holistic eradication or population control programs should consider these externalities to adequately and efficiently address their impacts.

## Introduction

Invasive species are mobile, have multiple vectors, and ignore property, jurisdictional, and tenure boundaries [1]. Effective strategies for reducing their populations require a comprehensive understanding of the factors that influence their presence and the damage they cause. Although reducing these populations can help mitigate these economic costs, restore

**Data availability statement:** Due to the NASS Confidentiality Pledge, the data underlying this article cannot be shared publicly. Secure access to NASS data may be obtained by agreement and sworn status only; restrictions apply. NASS allows eligible researchers the opportunity to gain access to restricted microdata files for select statistical research projects. NASS is responsible for protecting the confidentiality of its survey respondents. For this reason, prospective researchers commit to exclusively statistical uses of the data and data confidentiality must be considered before they are permitted access to NASS restricted data. Potential candidates must submit a research application. After receiving the application, NASS's Data Access and Disclosure Review Board (DADRB) will assess the proposal for feasibility based on several factors including fitness for use and potential disclosure risk. The Data Lab Handbook contains policies and regulations that all researchers are required to follow. The Nass Data Lab and Data Access Group can be contacted at SM.NASS.Data. Lab@usda.gov.

**Funding:** This research was supported by the U.S. Department of Agriculture, Animal and Plant Health Inspection Service and the National Feral Swine Damage Management Program. The funders had no role in study design, data collection and analysis, decision to publish, or preparation of the manuscript.

**Competing interests:** The authors have declared that no competing interests exist.

ecosystem balance, and preserve public health, the costs associated with public and private eradication and population control measures further add to the financial burden imposed by these invasive species. Additionally, the presence of invasive species creates spatial externalities that pose complex challenges for private landowners and public resource managers. [2] used the term *management mosaics* to refer to landscapes comprising many individually managed properties with a variety of uses. Research has shown that fragmented landscapes with diverse habitats and landowner management objectives can increase invasives species movement and spread, intensifying damage to agricultural lands [2–4].

Wild pigs (*Sus scrofa*: a.k.a. wild boar, wild/feral swine, wild/feral hogs [5]) are an invasive vertebrate species in the United States [6,7]. Despite a successful slowing of their expansion by the National Damage Management Feral Swine Program, the literature on the economic damages caused by wild pigs is growing [8,9]. Public and private efforts to reduce or eradicate these populations provide important benefits to society, including the mitigation of damages to agricultural operations [10,11], reducing the risk of disease transmission to domestic livestock or other wildlife [12–14], and the restoration of ecological balance following wild pig induced habitat degradation [15,16]. For instance, wild pigs cause large direct costs to crop and livestock operations each year from uprooting, feeding on, or trampling crops and pasture, preying on domesticated livestock, or causing other property damage like destroying fences or contaminating feed or water sources. Estimates of annual crop losses due to wild pig damage range upwards of several hundred million annually across the United States [11,17], and livestock losses may be even larger [14,18]. Wild pigs can also carry and transmit diseases to domestic livestock and other wildlife, further exacerbating economic losses and complicating disease management for livestock producers. For example, the spread of diseases such as African swine fever [19], foot-and-mouth disease [20] or leptospirosis [21] can result in economic damage and reduced economic activity along the livestock value chain from ranches to consumers. Wild pigs can also disrupt local ecosystems by competing with native wildlife, damaging natural habitats, and altering plant communities, soil composition, and water quality [22].

Two unaddressed potential determinants of the effectiveness of strategies for reducing wild pig damages to U.S. crop and livestock operations are the actions of neighboring landowners that attract wild pigs, whether inadvertently (establishing wildlife feed plots, neglecting fencing maintenance, etc.) or intentionally (transporting wild pigs for recreational hunting purposes), and the influence of landscape heterogeneity on wild pig behavior and distribution. For instance, while one landowner may implement measures to control wild pig populations, neighboring landowners may inadvertently attract them through practices such as maintaining water sources or other habitat features that create a refuge or intentionally for recreational hunting purposes [11], leading to a cumulative increase in pig presence. These actions complicate the ability to manage these populations effectively, as individual behavior can undermine broader control efforts. Moreover, differing perceptions of wild pigs where some landowners view them as pests while others see them as game animals, create tensions that hinder cooperative management efforts. Given these complexities, there is a need for information on the determinants of wild pig damages to effectively inform policy and management strategies that can mitigate their impacts. Addressing these factors can enhance our understanding of wild pig dynamics and improve targeted management interventions.

The objective of this research is to investigate how neighboring landowner actions and landscape mosaic factors affect the likelihood of wild pig presence and associated damages on U.S. farm and ranch operations. Our study contributes to the existing literature in three ways: First, we empirically examine the link between the actions of neighboring landowners and the incidence of wild pig presence and damage. Second, we analyze the role of landscape

heterogeneity or 'mosaic' factors, such as land use types and agricultural fragmentation, in either exacerbating or mitigating pig presence and damage. Third, we discuss the implications of accounting for these externalities in the design of management programs and policy. The findings will be valuable to academics, public agencies, and other stakeholder communities involved in wild pig management, as they underscore the need for a comprehensive understanding of both landowner behaviors and landscape characteristics due to these spatial externalities. Enhanced understanding of wild pig management could lead to more effective strategies to mitigate the economic damages caused by invasive wild pigs on U.S. crop and livestock operations.

The remainder of the article is organized as follows: In the next section, we outline the empirical strategy used to evaluate our hypotheses. We then describe the methods employed to collect and analyze the survey data. The results of our analysis are presented in the fourth section, followed by a discussion of the broader implications for wild pig management and policy.

## Materials and methods

### Model

We develop simple logistic regression specifications to model (1) the presence of wild pigs on the farm ( $P_i$ ) and (2) the existence of various damages resulting from wild pigs ( $D_i$ ),

$$P_i = f\left(A_i, L_i, N_i, X_i\right)$$

$$D_i = g\left(A_i, L_i, N_i, X_i\right)$$

where the vectors $A_i$, $L_i$, $N_i$, $X_i$, include respectively landowner $i$'s and their neighbors' wildlife attraction actions, landowner $i$'s property attributes, their neighbors' property attributes, and landscape characteristics describing the area where the farm is located. Logistic regression models help understand the influence of potential determinants on the likelihood of an occurrence (i.e., pig presence or damages) by calculating the probability of a binary outcome (yes/no) based on a set of explanatory variables. Each variable's coefficient indicates how much that variable increases or decreases the odds of the occurrence happening, indicating which have a significant impact on the predicted outcomes.

The dependent variable $P_i$ indicates the reported presence of wild pigs on the property in the previous year and is common to both the crop- and livestock-specific models. The dependent variable ( $D_i$ ) is defined separately according to the relevant questions in the crop and livestock surveys. "Pig crop damage" indicates the reported occurrence of some loss to crops (replant, crop damage, costly harvest) due to wild pigs. "Pig pasture damage" indicates damage to a livestock producer's pastureland. "Pig damage any" is a more broadly defined measure of damage from wild pigs. It indicates the presence of pig damage to any property or loss to crops for the crop survey, and pig damage to any property or damage to pastureland for the livestock survey. For the crop producers, crop damage regressions are performed separately at the property level and at the field level. We estimate seven individual models, four for crop producers and three for livestock producers.

Additionally, in response to the increasing damage and disease threats posed by wild pig populations in the US, the Feral Swine Eradication and Control Pilot Program (FSECPP) was established by the USDA's 2018 Farm Bill to provide additional resources in areas where feral swine have been identified as a particular threat, as determined by the Secretary. Farm Bill-Funded FSECPP counties, and those adjacent to FSECPP counties are depicted in S1 Fig in the

Supporting Information S1 File. To consider potential differences between wild pig presence and damages in FSECPP regions, we estimate these same seven logistic regressions only for operations in counties with Farm Bill-Funded FSECPPs and those adjacent to FSECPP counties. The 14 logistic regressions are summarized in S1 Table in S1 File.

## Data

To investigate the influence of these factors on pig presence and losses, we primarily use data from the 2021 survey of livestock producers (see [10]) and the 2022 survey of crop producers (see [11]). Both surveys were administered by the USDA National Agricultural Statistics Survey (USDA-NASS) [23] and designed by the USDA Animal and Plant Health Inspection Service-National Wildlife Research Center and Center for Epidemiology and Animal Health. The livestock survey included 13 U.S. states where invasive wild pigs were present in at least half of the counties in 2020: Alabama, Arkansas, California, Florida, Georgia, Louisiana, Mississippi, Missouri, North Carolina, Oklahoma, South Carolina, Tennessee, and Texas. The target population was livestock operations with pigs (*Sus domesticus*), cattle (*Bos taurus*), sheep (*Ovis aries*), or goat (*Capra hircus*) enterprises in the 13 states in 2020. The crop survey focused on the same states with the exclusion of Oklahoma and Tennessee, and targeted producers of corn (*Zea mays*), soybeans (*Glycine max*), wheat (*Triticum* spp.), rice (*Oryza sativa*), peanuts (*Arachis hypogaea*), and sorghum (*Sorghum bicolor*). Both surveys defined a wild pig as any pig or hog that is roaming free and does not have an owner. Other names provided to respondents as synonyms for wild pig for the purpose of responding to the study included feral hog, wild boar, wild pig, and feral swine.

The sections of particular interest in the survey are provided in the SI file and elicited information on (1) things that the respondent does or allows other people to do on their property, and things that are done on properties surrounding their operation (both surveys – S2 Fig in S1 File); (2) any cost incurred due to wild pig presence on pasture (livestock survey – S3 Fig in S1 File); (3) any cost incurred due to wild pig presence on field crops (crop survey – S4 and S5 Figs in S1 File); and (4) wild pig damage to property (both surveys – S6 Fig in S1 File).

The sampling procedure for both surveys followed a Multivariate Probability Proportionate to Size design. It resulted in sample sizes of 18,074 and 11,495 farm operations for the livestock and the crop surveys, respectively. The overall effective response rates (excluding refusals) were 44.5% for the livestock survey (8,035 respondents answering at least one question) and 39.4% for the crop survey (4,534 respondents). A complete description of the survey procedures for each survey and the NASS confidentiality considerations are provided in McKee et al. [10] for the livestock survey and in McKee, Mayer, Shwiff [11] for the crop survey.

For the crop survey, two datasets were created. The crop-level dataset (n = 4,484) includes one observation for each of the survey crops grown in the previous by the producers sampled, restricting the dataset to complete observations which information about the number of acres planted and the presence of any wild pig damage (crop loss, additional cost at replant and harvest) is available (S4 and S5 Figs in S1 File). There can be up to six observations per producer. It also includes answers to questions about measures taken on the property or surrounding properties (S2 Fig in S1 File). The property-level dataset (n = 2,426) was created by aggregating the first dataset at the producer level and includes additional information about property damage and the presence of wild pig on property in the previous year (see S6 Fig in the S1 File). For the livestock survey, a single dataset (n = 4,302) was created encompassing landowners that had pasture on their property during the previous year and that answered questions related to wild pig presence and damage, in the same 11 states as the crop survey.

The binary dependent variables that we construct from the survey data to estimate the logistic regression models are defined as wild pig presence (1 = yes, 0 = otherwise) and wild pig damages (1 = yes, 0 = otherwise). The population-weighted proportion of crop and livestock producers within each possible combination of values for these two dependent variables are shown in Fig 1. Wild pigs were present on over 40% of farms in our study area, with livestock farms reporting a slightly higher incidence of presence than crop farms. Surprisingly, there was a large correlation between wild pig presence and damages, with an overwhelming majority of those operations experiencing presence also experiencing damages. There were no discernible differences, however, in these patterns between farms in all counties and farms limited to those in FESCC and adjacent counties.

### Explanatory variables

The main explanatory variables in the model are also constructed using the USDA-NASS survey data (S2 Table in S1 File). A first set of binary variables represented by $A_i$ above indicate if the respondent potentially induced the presence of wild pigs by using feed to attract game animals on their property, and whether their neighbors engaged in activities that encouraged wildlife presence. The latter is set equal to one if the respondents' neighbors either used feed to attract game animals, engaged in recreational wild pigs on their property, or allowed others to do so for purposes other than control, transported and released wild pigs on their property, or more generally intentionally attracted or provided a "safe haven" for wild pigs (e.g., water or tree cover). The same variable is set to zero if the residents' neighbors took measures to control the presence of wild pigs.

A set of USDA-NASS survey-based variables make up the vector $L_i$ in the above equations. They include farm and operator exogenous characteristics such as any features on the respondent's property that may serve as a safe haven or otherwise attract wild pigs, a binary variable equal to one if the respondent's property size was greater than the median size in the sample, and a continuous variable indicating the acreage planted in crops or pastureland. For the livestock operations, we also included a series of binary variables indicating if the operation is a beef, dairy, hog, or sheep and goat farm. More details on the construction of these variables can be found in S2 Table in S1 File.

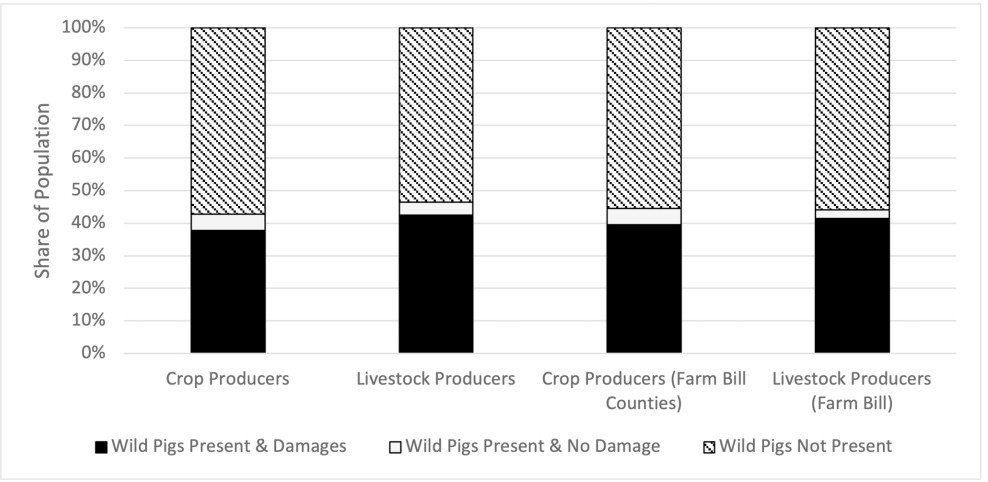

**Fig 1. Summary of wild pig presence and damages on U.S. Crop and Livestock farms.**

Two binary variables – the proximity to public parks, wildlife refuges, or other public land nearby that could serve as safe havens for wild pigs, and if surrounding properties employ farming or livestock ranching practices that may attract wild pigs, whether intentionally or unintentionally – make up the vector or neighboring property characteristics $N_i$.

We use the following definitions and data sources for other exogenous landscape variables represented by $L_i$. For landscape suitability we use a three-level expected equilibrium wild pig density factor variable based on levels: 0–3.99, 4–7.99, and 8–50 pigs/km². We created this latter variable based on wild pig densities by county in 2020 and 2021, which were generated using methods described in [24] by RS Miller (unpublished data). They are estimates of expected equilibrium wild pig abundance based on a combination of resources and other demographic processes such as hunting, assuming all available habitat is occupied. We will refer to them as "estimated equilibrium wild pig densities". For average agricultural property size in the county, we use data from the 2017 US Census of Agriculture. We did not include variables codifying the respondent's or the respondent's neighbors' wild pig management actions as the decision to engage in control actions is endogenous to the presence of wild pigs and the occurrence of wild pig damage. The relationship between management and the presence of damage is bidirectional: Wild pig management decreases the likelihood of damage, but the presence of damage increases the incentive to engage in management.

Table 1 presents summary statistics for the explanatory variables described above for the crop and livestock surveys. For the crop survey, summary statistics are presented at the property and field levels. About 43% of crop farmers across all surveyed counties observed wild pigs on their property at least once during 2020, while 38% experienced some monetary loss from wild pigs in that year. Damage to crops in the form of replanting, yield loss, or harvest costs occurred on 32% of crop farms. Counties targeted by Farm Bill funds or those adjacent to Farm Bill counties had slightly higher incidences of wild pig presence and damage (S3 Table in S1 File); wild pigs were present on 45% of crop farms in these counties vs. 34% with crop damage and 40% with any property damage from pigs. Incidence of pig damage was similarly high among livestock producers, with 46% of livestock farms reporting wild pigs, 36% experiencing damage to pastureland, and 42% experiencing any damage to pasture or property in 2021. Unlike crop farms, livestock producers in Farm Bill and Farm Bill-adjacent counties were slightly less impacted by wild pigs.

Feeding wildlife, which may potentially attract wild pigs, is similarly common among crop producers (32%) and livestock producers (31%). However, crop farmers are more likely to have neighbors that attract wild pigs than livestock farmers at 61% vs. 42%. About 60% of crop and livestock farms contain features that act as safe havens for pigs, while 21% and 15% of crop and livestock farms, respectively, are bordered by public lands that may be a haven for wild pigs.

The distribution of crop and livestock farms are similar in terms of estimated equilibrium wild pig densities. Over 85% of crop farms and over 91% of livestock farms are in counties with at least 4 pigs per km². Average farm sizes differ between the crop survey and the livestock survey. Crop producers operate in counties where the average farm is about 428 acres (roughly the U.S. average) compared to 160 acres for the livestock farms in our sample. Samples differ in the amount of agricultural land operated. Livestock operations have 341 acres of pastureland on average, while crop farms plant an average of 764 acres of cropland. The crop mix in our sample reflects the regional distribution of farms with the most growing corn, soybeans, or wheat. Among livestock producers, the vast majority raise beef cattle at 87% of operations. Domestic hogs, the next highest livestock type, are only raised on 4% of surveyed farms. Texas is the largest state represented in both samples, making up nearly half of all farms in the livestock survey and about 24% of crop farms in our sample.

**Table 1. Summary Statistics (All Counties).**

| Variables | | Crop Farms | | Crop Farms (Parcel) | | Livestock Farms | |
|---|---|---|---|---|---|---|---|
| | | Mean | Std. Error | Mean | Std. Error | Mean | Std. Error |
| Wild Pig Presence and Damage | | | | | | | |
| | Pig Presence | 0.428 | 0.018 | | | 0.464 | 0.018 |
| | Pig Damage to Crops or Pasture | 0.323 | 0.015 | | | 0.355 | 0.017 |
| | Pig Damage to Any Property | 0.378 | 0.016 | | | 0.424 | 0.18 |
| | Pig Damage to Crops (Field Level) | | | 0.259 | 0.012 | | |
| Attraction Actions (Landowner and/or neighbor) | | | | | | | |
| | Landowner Attracts Game Animals | 0.321 | 0.017 | 0.320 | 0.016 | 0.308 | 0.016 |
| | Neighbor Attracts Wildlife | 0.606 | 0.018 | 0.622 | 0.017 | 0.420 | 0.018 |
| Landowner Property Characteristics | | | | | | | |
| | Haven for Wild Pig | 0.592 | 0.019 | 0.607 | 0.018 | 0.620 | 0.020 |
| | Large Farm | 0.303 | 0.013 | 0.341 | 0.015 | 0.359 | 0.016 |
| | Crop or Pasture Acres (100s of acres) | 7.637 | 0.295 | 4.678 | 0.16 | 3.411 | 0.207 |
| | Beef Farm | | | | | 0.870 | 0.012 |
| | Dairy Farm | | | | | 0.029 | 0.005 |
| | Hog Farm | | | | | 0.043 | 0.006 |
| | Sheep or Goat Farm | | | | | 0.022 | 0.003 |
| | Corn | | | 0.331 | 0.011 | | |
| | Soybeans | | | 0.277 | 0.010 | | |
| | Wheat | | | 0.179 | 0.009 | | |
| | Rice | | | 0.059 | 0.004 | | |
| | Sorghum | | | 0.067 | 0.008 | | |
| | Peanuts | | | 0.086 | 0.007 | | |
| Neighboring Property Characteristics | | | | | | | |
| | Public Park, Wildlife Refuge, or Other Public Land Nearby | 0.212 | 0.015 | 0.222 | 0.015 | 0.153 | 0.012 |
| | Neighbor Farms | 0.466 | 0.019 | 0.462 | 0.018 | 0.410 | 0.018 |
| Landscape Characteristics | | | | | | | |
| | Carrying Capacity Low | 0.147 | 0.01 | 0.147 | 0.013 | 0.085 | 0.008 |
| | Carrying Capacity Medium | 0.705 | 0.015 | 0.709 | 0.016 | 0.742 | 0.018 |
| | Carrying Capacity High | 0.148 | 0.011 | 0.144 | 0.011 | 0.173 | 0.018 |
| | Avg. Farm Size (100s of Acres) | 4.279 | 0.139 | 4.678 | 0.160 | 1.596 | 0.074 |

**Notes**: Survey weighting applied using weights provided by USDA-NASS.

Of the 2,426 crop farms and 4,484 individual crop parcels in the full sample, 1,030 farms and 1,916 parcels belong to a FSECPP or FSECPP-adjacent county. For livestock operations, 1,404 out of 4,302 total farms are in a FSECPP or FSECPP-adjacent county. A similar table for the subset of farm bill counties can be found in S3 Table in S1 File.

## Results

We report the results of our logistic regressions in Tables 2 and 3. For ease of interpretation, we report only the average marginal effects for all models, which represent the estimated change in the likelihood of being impacted by wild pigs and are interpreted as percentage-point increases and decreases in the probability of being impacted by wild pigs. We are also careful to point out that while these estimates reveal useful associations between on-farm wild pig impacts and landowner, neighbor, and landscape characteristics, they are based on observational survey data and should not be interpreted as causal effects.

**Table 2. Marginal Effects of Determinants of Wild Pig Presence and Damage to Crop Operations.**

| | Pig Presence | | Pig Crop Damage | | Pig Damage Any | | Pig Crop Damage Parcel | |
|---|---|---|---|---|---|---|---|---|
| Variables | dy/dx[a] | Std. Error | dy/dx | Std. Error | dy/dx | Std. Error | dy/dx | Std. Error |
| Attraction Actions (Landowner and/or neighbor) | | | | | | | | |
| Landowner Attracts Game Animals | -0.024 | (0.028) | 0.005 | (0.027) | 0.004 | (0.030) | 0.005 | (0.022) |
| Neighbor Attracts Wildlife | 0.130*** | (0.027) | 0.123*** | (0.027) | 0.123*** | (0.029) | 0.110*** | (0.025) |
| Landowner Property Characteristics | | | | | | | | |
| Haven for Wild Pig | 0.169*** | (0.026) | 0.119*** | (0.027) | 0.150*** | (0.027) | 0.079*** | (0.024) |
| Large Farm | 0.113*** | (0.026) | 0.139*** | (0.024) | 0.135*** | (0.026) | 0.095*** | (0.021) |
| Crop or Pasture Acres (100s) | 0.002** | (0.001) | 0.003*** | (0.001) | 0.002** | (0.001) | 0.004*** | (0.001) |
| Soybeans | | | | | | | -0.072*** | (0.019) |
| Wheat | | | | | | | -0.039 | (0.028) |
| Rice | | | | | | | -0.033 | (0.034) |
| Sorghum | | | | | | | -0.037 | (0.046) |
| Peanuts | | | | | | | 0.074*** | (0.027) |
| Neighboring Property Characteristics | | | | | | | | |
| Public Park, Wildlife Refuge, or Other Public Land Nearby | 0.117*** | (0.028) | 0.099*** | (0.027) | 0.093*** | (0.029) | 0.091*** | (0.022) |
| Neighbor Farms | 0.089*** | (0.026) | 0.083*** | (0.024) | 0.105*** | (0.026) | 0.073*** | (0.021) |
| Landscape Characteristics | | | | | | | | |
| Carrying Capacity Medium | 0.168*** | (0.052) | 0.137*** | (0.045) | 0.172*** | (0.049) | 0.157*** | (0.036) |
| Carrying Capacity High | 0.151** | (0.061) | 0.124** | (0.053) | 0.154*** | (0.058) | 0.157*** | (0.036) |
| Avg. Farm Size (100s of Acres) | -0.007** | (0.004) | -0.006* | (0.003) | -0.007* | (0.004) | -0.007** | (0.003) |
| State Fixed Effects | Yes | | Yes | | Yes | | Yes | |
| Observations | 2,426 | | 2,426 | | 2,426 | | 4,484 | |
| Pseudo R2 | 0.553 | | 0.459 | | 0.516 | | 0.415 | |

**Notes**:

***p < 0.01,

**p < 0.05,

* p < 0.1.

[a]dy/dx represents the marginal effects of each explanatory variable on the predicted probability of a crop farm or crop parcel experiencing wild pig presence or damage. All marginal effects are average marginal effects. All marginal effects are weighted using weights provided by USDA-NASS.

We begin by explaining the crop model results, which can be found in Table 2. Neighboring property owners' attraction behaviors are positively and significantly related to the presence of wild pigs and damage to crops or farm property. A neighboring landowner engaging in one or more of the defined attraction behaviors raises the probability of a crop farmer encountering wild pigs on their property by 13 percentage points and the probability of experiencing damage to crops or property by 12 percentage points. However, wildlife attraction by crop farmers is not associated with either pig presence or damage to their own property. Where present, the significance of neighbor attraction behavior may be endogenous to wild pig outcomes. For example, neighboring landowners who offer recreational hunting of wild pigs on their property may do so in response to the opportunity created by the presence of pigs on the surveyed farm's property. Disentangling these forces would require a quasi-experimental design that is beyond the scope of this study.

Examining the landowner's property characteristics highlights that farms with natural safe havens for wild pigs are about 17 percentage points more likely to have pigs on their property than farms without these features. They are also more likely to experience damage from wild

**Table 3. Marginal Effects of Determinants of Wild Pig Presence and Damage to Livestock Operations.**

| | Pig Presence | | Pig Pasture Damage | | Pig Damage Any | |
|---|---|---|---|---|---|---|
| VARIABLES | dy/dx[a] | Std. Error | dy/dx | Std. Error | dy/dx | Std. Error |
| Attraction Actions (Landowner and/or neighbor) | | | | | | |
| Landowner Attracts Game Animals | 0.131*** | (0.025) | 0.079*** | (0.025) | 0.101*** | (0.025) |
| Neighbor Attracts Wildlife | 0.049* | (0.028) | 0.032 | (0.025) | 0.039 | (0.029) |
| Landowner Property Characteristics | | | | | | |
| Haven for Wild Pig | 0.171*** | (0.027) | 0.216*** | (0.024) | 0.211*** | (0.027) |
| Large Farm | 0.149*** | (0.024) | 0.148*** | (0.023) | 0.155*** | (0.024) |
| Crop or Pasture Acres (100s of acres) | 0.002** | (0.001) | 0.000 | (0.000) | 0.001 | (0.001) |
| Beef Farm | 0.105*** | (0.040) | 0.129*** | (0.046) | 0.107** | (0.043) |
| Dairy Farm | 0.126** | (0.052) | 0.144*** | (0.051) | 0.119** | (0.053) |
| Hog Farm | -0.069 | (0.048) | -0.073 | (0.059) | -0.057 | (0.050) |
| Sheep or Goat Farm | -0.119* | (0.066) | -0.096 | (0.061) | -0.073 | (0.063) |
| Neighboring Property Characteristics | | | | | | |
| Public Park, Wildlife Refuge, or Other Public Land Nearby | 0.082 | (0.051) | 0.049 | (0.034) | 0.080* | (0.048) |
| Neighbor Farms | 0.096*** | (0.029) | 0.079*** | (0.027) | 0.097*** | (0.029) |
| Landscape Characteristics | | | | | | |
| Carrying Capacity Medium | 0.121** | (0.056) | 0.134*** | (0.042) | 0.154*** | (0.051) |
| Carrying Capacity High | 0.134* | (0.070) | 0.176*** | (0.054) | 0.195*** | (0.068) |
| Avg. Farm Size (100s of Acres) | -0.009 | (0.007) | -0.008 | (0.006) | -0.005 | (0.006) |
| State Fixed Effects | Yes | | Yes | | Yes | |
| Observations | 4,302[a] | | 4,302 | | 4,302 | |
| Pseudo R2 | 0.560 | | 0.510 | | 0.53 | |

Notes:

***$p < 0.01$,

**$p < 0.05$,

* $p < 0.1$.

[a]dy/dx represents the marginal effects of each explanatory variable on the predicted probability of a livestock farms experiencing wild pig presence or damage. All marginal effects are average marginal effects. All marginal effects are weighted using weights provided by USDA-NASS.

pigs—the probability of damage to crops and all property are 12 and 15 percentage points higher, respectively, for farms with natural safe havens. Larger operations are similarly more likely to be affected by wild pigs. The probability of presence is about 11 percentage points higher among operations in the upper 50th percentile for total acres owned or rented than those in the bottom 50th percentile. The likelihood of damage to crops or any property due to wild pigs rises by about 14 percentage points for large farms. Controlling for the total amount of land operated, a 100-acre increase in crop acres planted is associated with a modest 0.2 percentage-point increase in the likelihood of encountering wild pigs at the property level and a 0.4 percentage-point increase at the parcel level. Changes in damage probabilities are similarly small, suggesting total land operated is more important for wild pigs than the acreage devoted to crop production. Compared to corn parcels, soybean and peanut parcels are respectively six percentage points less likely and 15 percentage-points more likely to be damaged.

Proximity to public land appears to matter for wild pig exposure. Farms that neighbor a public park, wildlife refuge, or other public lands are 12 percentage points more likely to have wild pigs on their property, and about 10 percentage points more likely to suffer some damage from pigs. The probability of pig presence and damage is also higher among neighboring farms. Bordering another farm is associated with a nine percentage-point increase in

the probability of wild pig presence, an eight percentage-point increase in the probability of incurring crop damage, and an 11 percentage-point increase in the probability of suffering any damage from wild pigs.

As expected, a county's estimated population density of wild pigs is positively related to the local incidence of wild pigs. Relative to counties with low estimated densities (less than four pigs/km$^2$), crop farmers in counties with moderate pig densities (4–7.99 pigs/km$^2$) are 17 percentage points more likely to have wild pigs on their property. The estimated difference in the probability of presence falls to 15 percentage points for counties with high pig densities (eight or more pigs/km$^2$). A similar pattern is observed for wild pig damage to crops and all property. However, the coefficients for high- and low-density counties are not statistically different, suggesting that pig density effects set in at around 4 pigs/km$^2$, but do not worsen beyond this threshold. Finally, average farm size in the county is weakly negatively related to on-farm impacts to crop operations. A 100-acre increase in the average farm size in the county is associated with a 0.7 percentage-point reduction in the probability of observing wild pigs on any individual operation. The estimated impact is similar for the probability of damage to crops and any property, though the estimated marginal effects are only significant at the 0.1 level.

Turning to the results for livestock producers (Table 3), we observe some differences in the importance of certain factors. Unlike crop producers, livestock producers that attract game animals are significantly more likely to have wild pigs on their operations, resulting in a greater likelihood of damage. Attraction efforts are associated with a 13 percentage-point increase in the likelihood of wild pig presence, an 8 percentage-point increase in the likelihood of damage to pasture, and a 10 percentage-point increase in the likelihood of any damage (including to pasture) on livestock farms. Where neighbor attraction behaviors are positively and significantly related to wild pig impacts on crop farmers, they are only weakly associated with the presence of wild pigs for livestock farms. The likelihood of presence on livestock operations rises by 5 percentage points when neighboring landowners engage in actions that attract wild pigs. The association is only significant at the 0.1 level. The estimated marginal effects of neighbor behavior on damage to livestock farms are similarly small but not statistically significant at any level.

The impact of livestock operation characteristics is generally as expected. Livestock properties that include natural safe havens for wild pigs are 17 percentage points more likely to host wild pigs than those without, and the probability of damage to pasture or any farm property rises by 21–22 percentage points. Livestock operations with large land holdings are also more likely to be impacted by wild pigs, with a 15 percentage-point higher probability of presence and 15–16 percentage-point higher probability of damage to pasture and any property. The amount of pasture acres is positively but weakly related to wild pig detection. Holding total land operated constant, a 100-acre increase in pastureland is associated with a 0.2 percentage-point increase in the likelihood of presence. The marginal effects of pasture acreage on damage from wild pigs are near zero and statistically insignificant.

Wild pig impacts also vary by livestock farm type. Operations with beef cattle are 11 percentage-points more likely to have wild pigs on their properties while those with dairy cattle are 13 percentage-points more likely. For beef and dairy producers, the likelihood of damage to pasture acres and property also rises by between 11 and 14 percentage points. Marginal effects for hog farms and sheep and goat farms are negative in all cases. In only one case—sheep and goat farms in the presence model—is the association statistically significant at the 0.1 significance level.

In contrast to crop farms, locating near public land that offers shelter for wild pigs does not make the presence of wild pigs on livestock operations more likely. Public land-adjacent

livestock farms are slightly more likely to experience some form of damage from a wild pig—increasing by an estimated eight percentage-points, all else equal—though the effect is weakly significant. Having a neighbor that farms, however, has a similar impact on both livestock and crop operations. The probability of presence rises by 10 percentage points for livestock operations that border other farms. Pig damage to pastureland and property also become more likely, rising 8 and 10 percentage points, respectively.

Finally, estimated equilibrium wild pig density predicts wild pig impacts on individual livestock farms. For livestock operations in counties with a medium, the probability of wild pig presence rises by 12 percentage points relative to a county with lower density. For high-density counties, the probability of presence increases by 13 percentage points. The likelihood of wild pigs damaging livestock pastureland rises by 13 and 18 percentage points for medium and high-density counties, respectively, while the likelihood of experiencing property damage from wild pigs grows by 15 and 20 percentage points. Though the sign and size of the marginal effects for average farm size are comparable to the crop models, they are not statistically significant for livestock operations at conventional levels.

The above relationships may be mitigated by public policy directed at wild pigs. Results for the FSECPP restricted sample are shown in S4 and S5 Tables in S1 File. Exposure to an FSECPP program may reduce the impacts of the factors associated with wild pig presence and damage. If so, the marginal effects discussed above will be attenuated toward zero and/or lose statistical significance. In general, we find that in most cases, the marginal effects for the full sample of crop and livestock farms shrink slightly and many become less statistically significant, suggesting that federal action may be effective for disrupting the vectors through which wild pigs affect landowners. A notable exception, however, is the estimated wild pig density of the county. For both the crop and livestock samples, the number of pigs/km$^2$ appears to have a larger impact on the presence of wild pigs on farms and the resulting damage to farm property in FSECPP and FSECPP -border counties than the broader sample. A potential explanation may be the criteria for selection into the FSECPP, which prioritized counties with large pig populations to begin with. We are careful to note that these differences in marginal effects between samples are small and may be driven by sample size, sample selection, or other unobserved factors that influence the strength of our estimates. Differences between results should be interpreted with this in mind.

## Discussion

This study provides several important insights. First, the behavior of landowners and neighbors influences the presence of wild pigs and likelihood of damage to farm property, with these relationships appearing to be heterogeneous across farm types. Wildlife attraction appears play a significant role in wild pig presence and the damage they cause to livestock operations, but not crop operations. Conversely, the attraction behavior of neighboring property owners is significant for crop operations but not livestock operations. This implies that crop farmers are vulnerable to externalities from neighboring landowners that attract wild pigs. Consequently, this study adds to the existing literature on local agricultural externalities (e.g., [24,25]) by providing empirical evidence that illustrates how wild pigs represent another example of such externalities. The findings also suggest that the role of cooperation among nearby landowners could be given greater consideration in future wild pig management efforts. Here, insight and lessons could be gained from related research on managing other types of invasive species and externalities [26–28].

Second, we found that certain characteristics of farms and nearby properties significantly influence the impact wild pigs have on agricultural landowners. Notably, the presence of water sources, forested areas, and/or low-lying brush within the farm raises the probability of wild

pigs being present and imposing damage to crops, pasture, or other property. Furthermore, farms that border public lands, such as parks or wildlife refuges, also experience a higher risk of wild pig-related impacts in a similar way, particularly among crop farms. Larger crop and livestock operations are more likely to encounter wild pigs. This may be due due to a detection effect where wild pigs are simply more present on larger operations, regardless of their specific characteristics. Among livestock operations, beef and dairy operations are the most impacted by wild pigs, likely because they often store hay, water resources, and other animal feed stored outdoors. This finding is important for preventing the spread of pig disease, because the public economic costs of damage caused by wild pigs could be significantly higher for the livestock sector due to the associated risk of disease transmission (e.g., [14]). These types of cost could extend beyond the private operations and include impacts on food security, food prices, and public health.

Our third main finding is that when we focus on counties with an FSECPP designation, and those adjacent to an FSECPP county, we find slight reductions in the magnitude and significance of most associations. This suggests some limited success of public control and eradication programs for farmers. However, a comparison of summary statistics does not show meaningful differences in the behavior of farmers or their neighbors in FSECPP and non-FSECPP counties. This aspect of our study merits further attention in future studies. As Duncan et al. [29] highlight in their study of the impacts of the Feral Swine Eradication and Control Pilot Program, developing an improved understanding of these program impacts can help inform the continuation or expansion of such efforts in future farm bills. This study therefore also contributes to the growing body of literature that explores wild pig policy and legislation (e.g., [30]).

Overall, our study supports the notion that wild pig management in the United States is influenced by the actions of neighboring landowners and the characteristics of the surrounding landscape. This research therefore emphasizes the necessity of considering spatial externalities in formulating effective management policies. It suggests that the impacts of wild pig management likely extend beyond the specific management area. Managing public lands, for example, can protect neighboring private lands. As wild pig populations continue to pose significant threats to agricultural productivity and ecological balance, targeted interventions that promote cooperation among landowners and acknowledge landscape variability are essential.

This study also had several limitations. First, due to significant under-reporting and a wide range of estimates, the dollar amount of monetary damages sustained by survey respondents could not be modeled. Given this, we cannot test how various attributes affect the overall cost of wild pigs to agricultural producers, and instead we use the discrete variable for whether damage occurred or not. Second, the heterogeneity in impacts we observe for some attributes between crop and livestock farms may be driven by unobserved differences in timing and/or implementation across farm types. Wildlife attraction behavior, for example, may be performed in the late fall after harvest, making the timing of the attraction behavior potentially different from the timing of most wild pig impacts, which may occur during the growing season. Third, to estimate the causal link between management and wild pig outcomes, a random experimental design where affected property owners are assigned to implement control behaviors, perhaps through a lottery system, would be required. Finally, our study's geographical scope was limited to the continental United States. However, analogous issues exist around the world, with other regions also experiencing problems related to crop and livestock damage caused by wild animals [31–33]. Future research that combines insights and evidence from previous studies on this topic from a global perspective would provide a valuable contribution to the literature by highlighting common challenges and potential solutions that could be applied across different contexts.

## Summary and conclusions

Wild pigs impose significant and growing costs on U.S. farm operations. This study examines differences in wild pig prevalence and damage incidence across farms in the southeastern United States using novel survey instruments of crop and livestock producers in 2020 and 2021. We examine how individual landowners' actions, their neighbors' behaviors, and landscape features influence the likelihood of wild pig damage on farms, allowing us to distinguish between direct effects and cross-boundary spillover impacts. Our results provide empirical evidence supporting spatial externality effects in wild pig management. We find a significant correlation between neighboring landowner actions and increases in wild pig presence and damage. Additionally, the likelihood of wild pig presence and damage increases as the diversity of neighboring land uses increases. This highlights the importance of considering social and landscape heterogeneity when devising management strategies.

These findings can inform policy design for current and future wild pig control programs directed at working agricultural lands, including those under the FSECPP. Greater understanding of the landowner, property, and landscape characteristics most associated with wild pig impacts can help tailor control and eradication efforts to specific areas and farm types. Specifically, this work underscores the need for more research into how to allocate limited resources for wild pig management based on property and landscape characteristics. The implications of this study extend beyond the agricultural realm, highlighting the importance of integrated management strategies that encompass both ecological and economic dimensions. Policymakers and land management agencies that prioritize collaborative frameworks to incentivize landowners to adopt practices that minimize wild pig attraction and enhance habitat management will likely lead to more effective and sustainable management strategies, reducing the economic burdens associated with wild pig damages while promoting ecological integrity and agricultural resilience.

## Supporting information

**S1 File.** Supporting Information. S1 Fig. Counties with Farm Bill-Funded Feral Swine Eradication and Control Pilot Programs (FSECPP) and those adjacent to FSECPP counties. Source: [25]. S2 Fig. Survey questions related to actions that attract wild animals on the respondent's property and on surrounding properties. (Source: [10,11]). S3 Fig. Survey questions related to wild pig damages on pasture. (Source: [10]). S4 Fig. Survey questions related to wild pig damages to cropland. (Source: [11]). S5 Fig. Survey questions related to wild pig damages to other property. (Source: [10,11]). S1 Table. Summary of 14 Logistic regression models evaluated. S2 Table. Description of the USDA NASS survey questions used to construct the study variables. S3 Table. Summary Statistics (Farm Bill Counties and Adjacent). S4 Table. Marginal Effects of Determinants of Feral Pig Presence and Damage to Crop Operations for Farm Bill and Farm Bill Adjacent Counties. S5 Table. Marginal Effects of Determinants of Feral Pig Presence and Damage to Livestock Operations for Farm Bill and Farm Bill Adjacent Counties. (DOCX)

## Acknowledgments

The authors would like to express their sincere gratitude to the National Feral Swine Damage Management Program for their support throughout this research. We also extend our thanks to the participants of the 2024 Western Agricultural Economics Association annual meeting for their insightful comments and suggestions, which significantly improved the quality of this work.

## Author contributions

**Conceptualization:** Sophie C. McKee, Nathan D. DeLay, Daniel F. Mooney.

**Data curation:** Sophie C. McKee, Nathan D. DeLay.

**Formal analysis:** Sophie C. McKee, Nathan D. DeLay, Daniel F. Mooney.

**Funding acquisition:** Stephanie A. Shwiff.

**Methodology:** Sophie C. McKee, Nathan D. DeLay, Daniel F. Mooney.

**Software:** Sophie C. McKee, Nathan D. DeLay.

**Writing – original draft:** Sophie C. McKee, Nathan D. DeLay, Daniel F. Mooney.

**Writing – review & editing:** Sophie C. McKee, Nathan D. DeLay, Daniel F. Mooney, Stephanie A. Shwiff.

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
