## [Decision Letter · Decision Letter 0]

14 Jan 2025

PONE-D-24-51915Externalities in wild pig damages on U.S. crop and livestock farms: The role of landowner actions and landscape heterogeneityPLOS ONE

Dear Dr. McKee,

Thank you for submitting your manuscript to PLOS ONE. After careful consideration, we feel that it has merit but does not fully meet PLOS ONE’s publication criteria as it currently stands. Therefore, we invite you to submit a revised version of the manuscript that addresses the points raised during the review process.

We look forward to receiving your revised manuscript.

Kind regards,

Muhammad Umer Arshad

Academic Editor

PLOS ONE

Journal Requirements:

2. Thank you for stating the following financial disclosure: [This research was supported by the U.S. Department of Agriculture, Animal and Plant Health Inspection Service and the National Feral Swine Damage Management Program].

3. Thank you for uploading your study's underlying data set. Unfortunately, the repository you have noted in your Data Availability statement does not qualify as an acceptable data repository according to PLOS's standards.

4. We notice that your supplementary figures are included in the manuscript file. Please remove them and upload them with the file type 'Supporting Information'. Please ensure that each Supporting Information file has a legend listed in the manuscript after the references list.

5. We note that Figure 1 in your submission contain [map/satellite] images which may be copyrighted. All PLOS content is published under the Creative Commons Attribution License (CC BY 4.0), which means that the manuscript, images, and Supporting Information files will be freely available online, and any third party is permitted to access, download, copy, distribute, and use these materials in any way, even commercially, with proper attribution. For these reasons, we cannot publish previously copyrighted maps or satellite images created using proprietary data, such as Google software (Google Maps, Street View, and Earth). For more information, see our copyright guidelines: http://journals.plos.org/plosone/s/licenses-and-copyright .

We recommend that you contact the original copyright holder with the Content Permission Form (http://journals.plos.org/plosone/s/file?id=7c09/content-permission-form.pdf ) and the following text:

“I request permission for the open-access journal PLOS ONE to publish XXX under the Creative Commons Attribution License (CCAL) CC BY 4.0 (http://creativecommons.org/licenses/by/4.0/ ). Please be aware that this license allows unrestricted use and distribution, even commercially, by third parties. Please reply and provide explicit written permission to publish XXX under a CC BY license and complete the attached form.”

Additional Editor Comments:

The overall paper "economic and ecological impacts of wild pig damages on U.S. agriculture" The study is robust and has significant policy implications; however, there are several areas that require further attention to enhance the paper's rigor and clarity

Here are few suggestions need to be considered

1. potential endogeneity in the variables capturing neighbor behavior should be addressed

2. restricted access to USDA survey data limits reproducibility. Providing detailed guidance on how future researchers might access or replicate the data using publicly available alternatives would improve transparency.

3. While the logistic regression models are appropriate for exploring associations, the limitations of observational data in establishing causality should be explicitly discussed.

4.The discussion section should include a more detailed comparison of the findings with prior studies to contextualize the results

5. The conclusion should be concise and focused on key findings and policy implications.

Reviewers' comments:

Reviewer's Responses to Questions

**Comments to the Author**

1. Is the manuscript technically sound, and do the data support the conclusions?

Reviewer #1: Partly

Reviewer #2: Yes

2. Has the statistical analysis been performed appropriately and rigorously? 

Reviewer #1: Yes

Reviewer #2: Yes

3. Have the authors made all data underlying the findings in their manuscript fully available?

Reviewer #1: Yes

Reviewer #2: Yes

4. Is the manuscript presented in an intelligible fashion and written in standard English?

Reviewer #1: Yes

Reviewer #2: Yes

5. Review Comments to the Author

Reviewer #1: Dear Authors,

The research work is interesting in its related fields. I have looked at it carefully, and it should be modified before publication process.

Introduction part line 105; you add your results in the inappropriate place; please paraphrase or transfer to the results part.

How did you analyze your collected data? It should be clear and simple for understanding. Lacks clarity your method of analysis

there is no consistency in using % or percentile; please use either of these throughout your text body.

In tables, there is dy/dx; this symbol/formula, needs what it shows in the table captions!

There is no discussion part, which is basic to relate your findings with others scholars?

Reviewer #2: Background

My field of expertise lies in farming, criminology, and penology. However, I served on the Predation Management Forum of South Africa for over 15 years as a representative of livestock farmers. This role provided me with extensive exposure to damage-causing animals and related issues, particularly in managing predation risks and fostering coexistence strategies. This experience has equipped me with a unique interdisciplinary perspective on the challenges and dynamics of human-wildlife interactions.

Through this lens, I accepted the request to peer review this paper. My background enables me to critically evaluate the research while appreciating the practical implications of its findings, especially for those engaged in wildlife management and agricultural operations.

Summary of the research and my impressions

This paper significantly contributes to the academic discourse on human-wildlife conflict by addressing the growing challenge of wild pig prevalence and its impact on agricultural operations in the southeastern United States. The research offers valuable insights into the interplay of landowner behaviour, neighbouring property actions, and landscape features, shedding light on the dynamics that influence wild pig impacts. The study’s findings on spatial externalities, particularly the roles of public lands and cross-boundary behaviours, are especially relevant for policymakers seeking to develop collaborative management strategies.

The paper is methodologically robust, presenting a straightforward research question and hypothesis supported by well-structured arguments and comprehensive survey data. Its exploration of landowner behaviour and neighbouring actions is insightful, providing a nuanced understanding of how individual and collective actions affect wild pig management. These findings have practical implications for designing targeted strategies to mitigate the economic and environmental damage caused by wild pigs.

Despite these strengths, there are areas where the paper could be refined to enhance clarity, accessibility, and global relevance. For instance:

1. Clarification of Citations: On page 4, line 91, the paper references "research" without specifying which studies are cited. Providing detailed citations in such instances would bolster the paper’s credibility and situate its findings within the broader body of literature.

2. Introduction of Key Concepts: The concept of wild pigs as res nullius animals (not owned by any individual) is introduced relatively late in the paper, on page 7. Including this concept in the introduction would provide essential context and help frame the issue for readers early in the discussion.

3. Global Perspective: While the paper focuses on the southeastern United States, it misses an opportunity to connect its findings to similar challenges faced in other parts of the world. Human-wildlife conflicts involving res nullius animals are a global issue, and a brief reference to analogous situations in other regions could broaden the paper’s applicability. Comparative research, even in summary form, would highlight shared challenges and potential cross-contextual solutions, enriching the discussion.

Addressing these areas would enhance the paper’s clarity and relevance to a wider audience, particularly those working on wildlife management in diverse ecological and socio-economic contexts. Furthermore, such improvements would make the paper more accessible to an international readership while emphasising the broader significance of its findings.

In conclusion, this paper is valuable to studying human-wildlife conflict and offers practical insights for managing wild pig impacts in agricultural settings. Minor amendments to improve clarity, citation practices, and global contextualisation would achieve greater impact and utility. I recommend it for publication, with attention to these suggested refinements.

6. PLOS authors have the option to publish the peer review history of their article (what does this mean? ). If published, this will include your full peer review and any attached files.

**Do you want your identity to be public for this peer review?** For information about this choice, including consent withdrawal, please see our Privacy Policy .

Reviewer #1: No

Reviewer #2: No

---

## [Author Response · Author response to Decision Letter 1]

14 Feb 2025

Please refer to the cover letter.

---

## [Editor Report · Decision Letter 1]

18 Feb 2025

Externalities in wild pig damages on U.S. crop and livestock farms: The role of landowner actions and landscape heterogeneity

PONE-D-24-51915R1

Dear Dr. McKee,

We’re pleased to inform you that your manuscript has been judged scientifically suitable for publication and will be formally accepted for publication once it meets all outstanding technical requirements.

Kind regards,

Muhammad Umer Arshad

Academic Editor

PLOS ONE
---

## [Editor Report · Acceptance letter]

PONE-D-24-51915R1

PLOS ONE

Dear Dr. McKee,

I'm pleased to inform you that your manuscript has been deemed suitable for publication in PLOS ONE. Congratulations! Your manuscript is now being handed over to our production team.

Kind regards,

on behalf of

Dr. Muhammad Umer Arshad

Academic Editor

PLOS ONE